# Lesion-Symptom Mapping of Acute Speech Deficits After Left vs. Right Hemisphere Stroke: A Retrospective Analysis of NIHSS Best Language Scores and Clinical Neuroimaging

**DOI:** 10.3390/brainsci15121329

**Published:** 2025-12-13

**Authors:** Nilofar Sherzad, Roger Newman-Norlund, John Absher, Leonardo Bonilha, Christopher Rorden, Julius Fridriksson, Sigfus Kristinsson

**Affiliations:** 1Arnold School of Public Health, University of South Carolina, Columbia, SC 29208, USA; nilofar40429@gmail.com; 2Department of Psychology, University of South Carolina, Columbia, SC 29208, USA; rnorlund@mailbox.sc.edu (R.N.-N.); rorden@mailbox.sc.edu (C.R.); 3School of Medicine, University of South Carolina, Greenville, SC 29605, USA; absher@mailbox.sc.edu; 4Department of Neurology, University of South Carolina, Columbia, SC 29201, USA; leonardo.bonilha@uscmed.sc.edu; 5Department of Communication Sciences and Disorders, University of South Carolina, Columbia, SC 29208, USA; fridriks@mailbox.sc.edu

**Keywords:** acute aphasia, neuroimaging, lesion-symptom mapping

## Abstract

**Background**: Recent research suggests that damage to right hemisphere regions homotopic to the left hemisphere language network affects language abilities to a greater extent than previously thought. However, few studies have investigated acute disruption of language after lesion to the right hemisphere. Here, we examined lesion correlates of acute speech deficits following left and right hemisphere ischemic stroke to clarify the neural architecture underlying early language dysfunction. **Methods**: We retrospectively analyzed 410 patients (225 left, 185 right hemisphere lesions) from the Stroke Outcome Optimization Project dataset. Presence and severity of speech deficits was measured using the National Institute of Health Stroke Scale Best Language subscore within 48 h of onset. Manual lesion masks were derived from clinical MRI scans and normalized to MNI space. Lesion-symptom mapping was conducted using voxelwise and region-of-interest analyses with permutation correction (5000 iterations; *p* < 0.05), controlling for total lesion volume. **Results**: Speech deficits were observed in 53.7% of the cohort (58.2% left, 48.1% right hemisphere lesions). In the full sample, the presence of speech deficits was associated with bilateral subcortical and perisylvian damage, including the external and internal capsules, insula, putamen, and superior fronto-occipital fasciculus. Severity of speech deficits localized predominantly to left hemisphere structures, with peak associations in the external capsule (Z = 6.39), posterior insula (Z = 5.64), and inferior fronto-occipital fasciculus (Z = 5.43). In the right hemisphere cohort, the presence and severity of speech deficits were linked to homologous regions, including the posterior insula (Z = 3.70) and external capsule (Z = 3.63), although with smaller effect sizes relative to the left hemisphere cohort. Right hemisphere lesions resulted in milder deficits despite larger lesion volumes compared with left hemisphere lesions. **Conclusions**: Acute speech impairment following right hemisphere stroke is associated with damage to a homotopic network encompassing perisylvian cortical and subcortical regions analogous to the dominant left hemisphere language network. These findings demonstrate that damage to the right hemisphere consistently results in acute speech deficits, challenging the traditional left-centric view of post-stroke speech impairment. These results have important implications for models of bilateral language representation and the neuroplastic mechanisms supporting language recovery.

## 1. Introduction

Aphasia is a debilitating language disorder that affects 20–40% of stroke survivors [1,2]. Although post-stroke speech impairment is commonly associated with damage to the perisylvian language network of the left hemisphere (LH), recent work suggests that damage to homotopic regions in the right hemisphere (RH) affects language abilities to a greater extent than previously thought [3,4,5]. However, few studies have explicitly investigated the association between RH lesion and acute disruption of language. To this end, the current study examined the relationship between LH/RH lesion location and presence as well as severity of speech deficits in a large cohort of acute ischemic stroke cases.

Concurrent models of speech/language processing provide a neurobiological framework underpinning cortical language networks, positing two major pathways: a left-lateralized dorsal stream supporting speech production and phonological processing, and a more bilaterally distributed ventral stream supporting speech comprehension and semantic processing [6,7,8]. Lesions affecting the dorsal stream are commonly associated with nonfluent speech and impaired verbal output, whereas lesions to the ventral stream more frequently disrupt speech comprehension [9,10,11,12]. While intersecting lines of research provide ample evidence for the importance of the LH for language processing, the contribution of the RH in linguistic processes and, specifically, language recovery, remains controversial.

The homotopic RH language network is engaged in various linguistic tasks in healthy individuals [13,14], including higher-order processes such as prosody, discourse, and pragmatic language use [15,16,17], as well as computational mapping of phonological [16,18,19] and semantic representations [20,21,22]. Prior work similarly indicates that right lateralized activity is increased during challenging language processing, suggesting that the RH supports linguistic processes in response to increasing task demands [23,24,25]. Moreover, in cases of extensive LH lesions at or shortly after birth, the contralateral RH language network can support language acquisition without any discernible behavioral differences [26,27]. This converging evidence has been taken to support the notion of a RH “weak shadow” homotopic network, reflecting nodes in a bilateral, albeit left-dominant, language processing network [26,28].

Although the RH language network is involved in linguistic processing in healthy individuals, the extent to which it can support neuroplastic functional reorganization of speech/language after stroke remains debated. The debate dates back to the late 19th century, when Thomas Barlow [29] described the case of a 10-year-old patient who recovered from aphasia after a stroke to the LH inferior frontal gyrus and later presented with similar language impairment following damage to the contralateral region in the RH. Other reports of sequential left/right hemisphere strokes [30,31,32] and language disruption during RH anaesthesia or virtual lesion [19,33,34,35,36], have been interpreted as evidence for the notion that the RH language network may compensate for language impairment caused by a LH stroke. Numerous prior studies have tested this hypothesis, yielding mixed results: some suggest compensatory RH involvement [37,38,39,40,41,42], whereas others indicate that its involvement is maladaptive for language recovery [41,43,44,45,46]. Correspondingly, recent meta-analyses highlight that some individuals with aphasia demonstrate recruitment of the RH language network, although it remains unclear which brain regions are implicated and whether they facilitate or deter language recovery [47,48,49].

Lesion-symptom mapping (LSM) studies offer a unique opportunity to examine the causal relation between damage to specific brain regions and speech impairment. Despite the debated contribution of the RH language network in language recovery in aphasia, few have explicitly investigated speech/language outcomes in patients with RH strokes in the absence of LH damage. Several recent publications have found that RH lesions are consistently associated with subtle impairments in chronic stroke survivors, though they rarely lead to classic aphasic syndromes [50,51,52,53,54,55]. While these findings suggest an independent contribution of the RH language network in speech/language processing, time-dependent functional reorganization within the intact dominant LH language network likely masks more subtle lesion-symptom associations.

LSM in acute patients is better suited to elucidate the immediate effects of damage to the RH language network on speech impairment [56,57]. Several prior studies indicate that acute speech deficits are commonly associated with damage affecting RH regions homotopic to canonical language regions in the LH, although patients tend to demonstrate greater overall recovery [2,4,58,59,60,61]. However, prior research has generally reported relatively small samples of participants [2], excluded patients with RH damage [12,62], or leveraged retrospective data from selected patients with RH damage and speech deficits [3,58,59], which results in an incomplete understanding of the regional distribution of lesion anatomy associated with speech deficits after right lateralized lesion. Better understanding of acute lesion anatomy related to RH speech impairment is critical to identify regions that contribute to linguistic processes, and to dissociate regions that may facilitate or deter language recovery in individuals with LH damage [28,42].

To this end, the current study aimed to characterize the relationship between LH/RH damage and presence as well as severity of speech deficits in a cohort of ischemic stroke cases from a large comprehensive stroke center. We hypothesized that severe deficits would be associated with extensive LH damage, particularly to classical language regions, while mild speech deficits would involve more heterogeneous lesions, including structures within the weak shadow RH language network. By clarifying the lesion correlates of verbal output deficits, this study aims to inform mechanisms of language recovery and support the development of personalized rehabilitation strategies in post-stroke aphasia.

## 2. Methods

### 2.1. Participants

This study used retrospective data from the Stroke Outcome Optimization Project (SOOP) [63]. Briefly, the SOOP dataset includes de-identified demographic data, functional assessments, and clinical neuroimaging data for 1106 individuals with acute ischemic stroke admitted to Prisma Health-Upstate hospital in Greenville, South Carolina, between January 2019 and December 2020 [63]. For the purpose of this study, a total of 410 eligible participants with available National Institute of Health Stroke Scale (NIHSS) Best Language subscores, demographic information, and neuroimaging data were included in data analyses. This retrospective evaluation of SOOP data was approved by the local ethics boards and Institutional Review Board (IRB) approval for data collection was obtained from Prisma Health Committee A, Greenville, South Carolina (Pro00078716, 10-29-2018). Participants’ characteristics are presented in Table 1.

### 2.2. Assessment of Speech Deficits

Participants were assessed with the NIHSS within 48 h of stroke onset. The NIHSS is a widely used clinical instrument for quantifying neurological impairment following stroke. It measures multiple functional domains, including consciousness, vision, facial palsy, motor strength, sensation, coordination, language, and speech [64,65]. The total score ranges from 0 to 42, with higher scores indicating more severe deficits. The NIHSS is routinely administered in both clinical and research settings to stratify stroke severity, monitor changes in neurological status, and predict functional outcomes [66,67,68,69].

The Best Language subtest of the NIHSS is designed to assess both expressive and receptive language function and to screen for the presence of aphasia in acute stroke patients. This item includes structured elicitation procedures involving naming, reading, and comprehension tasks, administered using a standardized stimulus sheet. First, the patient is asked to name six visually presented line drawings of common objects (e.g., glove, key, and cactus), probing lexical retrieval and expressive naming. Next, the patient reads aloud a simple sentence (e.g., “You know how.”), providing information about oral reading ability and articulatory fluency. Finally, receptive language is assessed by asking the patient to follow verbal commands (e.g., “Close your eyes” or “Make a fist”) to evaluate comprehension of spoken language.

Scoring of the Best Language subtest ranges from 0 to 3 and reflects the clinician’s judgment of the patient’s overall speech/language ability. A score of 0 denotes normal language function with no detectable aphasia. A score of 1 reflects mild to moderate speech deficits, in which the patient may have word-finding difficulties or mild comprehension deficits but can still communicate effectively. A score of 2 indicates severe speech impairment, with marked limitations in both comprehension and expression. A score of 3 denotes global aphasia or mutism, where the patient is unable to produce or comprehend language. Importantly, examiners are instructed to differentiate aphasia from other impairments that may interfere with performance (e.g., dysarthria or visual deficits) and to base scoring specifically on language function [70].

The Best Language subscore is a brief screening tool for speech impairment in acute stroke patients. Although it does not provide a detailed evaluation of aphasic language impairment, it is a widely used assessment to gauge the presence and severity of speech impairment in acute clinical settings [2,71,72]. Critically for the current study, its utility in predicting stroke-related outcomes, including lesion anatomy and functional recovery, has been previously demonstrated [71,73,74,75,76].

### 2.3. Neuroimaging Data Acquisition

MRI scans were acquired within 30 days of participants’ hospital admission, with the majority obtained within the first 48 h post-stroke. For each participant, the neuroimaging sequence offering the most complete brain coverage and optimal signal-to-noise ratio was selected. These typically included T1-weighted (T1w), T2-weighted (T2w), Fluid Attenuated Inversion Recovery (FLAIR), and diffusion-weighted imaging (DWI) sequences. The DWI scans comprised both TRACE images and those approximating apparent diffusion coefficient (ADC) contrast. Due to the clinical nature of data collection, scan acquisition parameters varied considerably across individuals. Sequence-specific details were preserved in the accompanying Brain Imaging Data Structure (BIDS) format sidecar files provided with each NIfTI image. T1w sequences, in particular, exhibited substantial variation in resolution, coverage, and contrast. Notably, many T1w scans included gadolinium contrast, which introduces properties distinct from the unenhanced scans typically used in research settings [63].

Raw DICOM files were converted to NIfTI format using dcm2niix [77]. To ensure participant anonymity, we applied a modified version of the spm_deface script from SPM12 to remove facial and neck features while preserving scalp anatomy [78]. We intentionally retained extracerebral tissue to support future modeling of image inhomogeneity and facilitate the development of brain extraction tools trained on clinically diverse datasets, including those with structural atypicalities such as post-surgical depressions and prominent diploic spaces.

### 2.4. Lesion Segmentation

Manual lesion segmentation was conducted to delineate stroke-affected tissue. Manual tracing of lesion boundaries on each axial slice of the native-space T2w image was performed by trained neuroscientists. Lesions were primarily identified using ADC maps, where acute stroke lesions typically appeared hypointense. Raters used MRIcroGL [79] to scroll through the ADC images and mark regions of unequivocal tissue injury. Contiguous lesion areas were traced across adjacent slices in the superior-inferior plane. Binary lesion masks were generated in native DWI space, with lesion voxels coded as ‘1’ and non-lesioned voxels as ‘0’. Each lesion mask matched the spatial resolution and dimensions of the original DWI image and was subsequently normalized to standard MNI space to support group-level analyses and anatomical labeling. In addition to acute lesion identification, participants were also evaluated for evidence of chronic infarcts visible on ADC scans. When present, chronic lesions were separately labeled, and corresponding binary masks were stored independently from acute lesion files. All lesion segmentation sessions were screen-recorded using QuickTime (v10.6.3) and can be made available upon reasonable request to the corresponding author.

### 2.5. Data Analysis

First, we performed voxelwise and region-of-interest (ROI) whole-brain analyses to identify lesion anatomy associated with (1) presence of speech deficits (NIHSS Best Language = 0 vs. ≥1) and (2) severity of speech deficits (ordinal NIHSS Best Language score: 0-3). Since only 6 participants presented with global aphasia, scores of ‘2’ and ‘3’ were collated and considered indicative of severe speech impairment in statistical analyses. Second, the same set of analyses was repeated separately in participants with lesion anatomy primarily restricted to the left (N = 225) and right (N = 185) hemispheres. Lesion-symptom mapping was performed using the NiiStat software (9-October-2016) and analyses included voxels or regions damaged in at least 10 participants. ROI-based analyses were guided by an anatomical parcellation developed by Faria et al. [80]. Overall lesion volume was treated as a covariate in all analyses, and permutation thresholding was applied to correct for multiple comparisons (N = 5000, corrected *p* < 0.05).

## 3. Results

Sample characteristics are presented in Table 1 (Appendix A). The full sample of participants had a mean age at stroke onset of 65.5 years (SD = 14.6), lesion volume of 38.7 cc (SD = 72.9), and NIHSS score of 8.2 (SD = 8.1). Of the 410 participants, 220 (53.7%) presented with speech deficits (NIHSS Best Language score ≥ 1). Relative to participants without speech deficits, participants with speech deficits presented with a larger lesion volume (48.6 vs. 27.3 cc, t = 3.01, *p* = 0.003), higher age (67.2 vs. 64.1 years, t = 2.09, *p* = 0.038), and higher overall NIHSS score (11.8 vs. 4.1, t = 11.2, *p* < 0.001). No group difference was observed for the distribution of sex (*χ*^2^ = 0.00, *p* = 1.00) or race (*χ*^2^ = 1.47, *p* = 0.230). Lesion overlap maps for participants with LH and RH lesions are presented in Figure 1.

### 3.1. Whole-Brain Analyses

We first ran whole-brain LSM analyses to determine lesion location associated with the presence and severity of speech deficits irrespective of which hemisphere was affected. ROI-based analysis revealed greater lesion load associated with the presence of speech deficits in bilateral subcortical, white matter, and insular structures, including left internal capsule (Z = 3.22), external capsule (Z = 3.22), putamen (Z = 3.16), and superior fronto-occipital fasciculus (Z = 2.91), and right insula (Z = 3.19) and external capsule (Z = 3.16). Voxelwise analysis yielded converging results, with the most robust association observed in a large cluster extending from the right insula (peak Z = 5.1) to pre- and postcentral gyri, internal and external capsules, and superior temporal gyrus. Smaller clusters were observed in left external and internal capsule, and right superior temporal gyrus.

In contrast, the severity of speech impairment (ordinal NIHSS Best Language score: 0, 1, 2–3) was associated with lesion load in left lateralized regions only, including subcortical regions, short and long white matter pathways, and cortical language areas (e.g., inferior frontal gyrus, superior temporal gyrus; Z = 3.31 to 6.39). Voxelwise analysis identified a large frontoparietal cluster extending into insular and basal ganglia structures (k = 10,875, peak Z = 6.8), and smaller cluster peaks in the superior corona radiata (Z = 5.3), precentral gyrus (Z = 5.1), and caudate nucleus (Z = 5.0). Figure 2 and Figure 3 visualize the multiple comparison-corrected ROI-based and voxelwise lesion maps, respectively, and effect sizes and regional distributions are presented in Appendix A.

### 3.2. Left Hemisphere LSM

The same analyses were run separately in patients with primarily left (N = 225) and right (N = 185) hemisphere lesions, respectively. A total of 58.2% of patients with LH lesion presented with speech deficits; 28.8% mild (1) and 29.3% severe (≥2; Figure 4). Relative to patients without speech deficits, patients with speech deficits presented with significantly greater LH lesion load (0.02 vs. 0.05, *p* = 0.004) and higher overall NIHSS score (4.5 vs. 12.2, *p* < 0.001).

Presence of speech deficits was strongly associated with lesion to deep subcortical and periventricular white matter structures, including the external capsule (Z = 3.77), superior corona radiata (Z = 3.68), and posterior limb of the internal capsule (Z = 3.68). Voxelwise analysis revealed peak clusters in the anterior limb of the internal capsule (k = 67, Z = 4.4) and external capsule (k = 53, Z = 4.3). Significant effects extended into the putamen (Z = 3.47) and the caudate nucleus (Z = 3.26), as well as inferior (Z = 3.51) and superior (Z = 3.41) fronto-occipital fasciculi, and posterior insula (Z = 2.84).

ROI-based and voxelwise LSM of speech impairment severity revealed similar regional distribution, with the most robust association observed in the external capsule (Z = 6.07) and extending into the insula, pre- and postcentral gyrus, inferior fronto-occipital fasciculus, and inferior frontal gyrus pars opercularis (k = 8323, peak Z = 6.4). Significant effects were also observed in the superior fronto-occipital (Z = 3.83) and uncinate (Z = 3.51) fasciculi, as well as the pole of the superior temporal gyrus (Z = 3.31). The full results are presented in Figure 2 and Figure 3 and Appendix A.

### 3.3. Right Hemisphere LSM

A total of 48.1% of patients with RH lesion presented with speech deficits; 34.6% mild (1) and 13.5% severe (≥2; Figure 4). A chi-square test revealed that significantly fewer patients presented with severe speech impairment following RH vs. LH lesion (*χ^2^* = 14.71, *p* < 0.001). Relative to RH patients without speech deficits, patients with speech deficits exhibited significantly larger lesion volume (24.7 vs. 61.6 cc, *p* < 0.001), greater lesion load (0.02 vs. 0.06, *p* < 0.001), higher overall NIHSS score (3.6 vs. 11.3, *p* < 0.001), and a larger proportion of African American patients presented with speech impairment compared to White patients (64.3 vs. 43.7%, *p* = 0.020).

Presence of speech deficits in RH patients was associated with similar lesion distribution as in LH patients, albeit with somewhat smaller effect sizes. ROI-based analysis revealed significant effects in the posterior insula (Z = 3.70), external capsule (Z = 3.63), insula (Z = 3.09), and the retrolenticular part of the internal capsule (Z = 3.02). Voxelwise analysis identified a large perisylvian cluster encompassing fronto-parietal cortical and subcortical structures, superior temporal gyrus, and peak in the insula (k = 24,800, peak Z = 6.3). Smaller clusters were observed in the supramarginal gyrus (Z = 4.9), superior temporal gyrus (Z = 4.7), and precentral gyrus (Z = 4.5).

Less robust findings were observed for speech impairment severity. Two regions survived permutation thresholding in the ROI-based analysis: external capsule (Z = 3.59) and posterior insula (Z = 3.46). Correspondingly, a single small cluster with a peak in the insula extending into the external capsule and inferior fronto-occipital fasciculus was identified in the voxelwise analysis (k = 130, peak Z = 5.5).

## 4. Discussion

In this study, we leveraged clinical neuroimaging and NIHSS Best Language scores to identify lesion patterns associated with the presence and severity of acute post-stroke speech deficits following damage to the left or right hemispheres. Our results revealed that the presence of acute speech deficits at stroke onset was associated with lesion to deep white matter structures, long association pathways, and perisylvian cortical areas in the left and right hemisphere. Consistent with our hypotheses, damage to perisylvian cortex and adjacent white matter regions in the left hemisphere was strongly associated with more severe speech impairment, whereas milder impairment was observed in patients with damage affecting homolog areas in the right hemisphere. Prior work in this area has been constrained by small sample sizes, exclusion of RH patients, or focus on chronic recovery phases where compensatory reorganization may mask effects. Thus, our large bilateral cohort (N = 410; 185 RH) assessed within 48 h of onset represents one of the largest acute LSM investigations of RH speech deficits to date, providing unprecedented power to characterize bilateral lesion-symptom associations before time-dependent reorganization. The implications of these findings are discussed below.

### 4.1. Presence vs. Severity of Speech Deficits

Presence of acute speech deficits (NIHSS Best Language ≥ 1) mapped onto bilateral subcortical and perisylvian networks, including the external capsule, internal capsule, putamen, insula, and superior fronto-occipital fasciculus. The strongest effect was observed in the bilateral external capsule, reflecting acute disruption of ventral association fibers connecting frontal, temporal, and insular cortices critical for semantic-phonologic integration [12,81,82,83]. Damage to the bilateral internal capsule was also consistently associated with the presence of speech deficits. Acute internal capsule lesion affects corticobulbar and thalamocortical projection fibers involved in articulatory feedback control [83,84,85]. Lesions to the insula and putamen have similarly been associated with impaired speech-motor planning and sequencing, often resulting in apraxic or dysarthric speech [84,86,87,88]. Involvement of the superior fronto-occipital fasciculus further suggests that acute speech impairment emerges from disruption of structural connections between anterior and posterior nodes within the language network [89,90,91]. The bilateral pattern of lesion anatomy observed in this sample indicates that damage to RH regions homologous to speech/language regions in the dominant LH is more consistently associated with the presence of acute speech deficits than previously thought [2,58,60].

In contrast, severity of acute speech impairment was predominantly associated with lesion to multiple LH structures, including basal ganglia (e.g., putamen, caudate nucleus), deep white matter projection and association fibers (e.g., internal/external capsule), long association pathways (e.g., inferior fronto-occipital fasciculus), and cortical language areas (e.g., inferior frontal gyrus pars opercularis, superior temporal gyrus). The predominance of subcortical over cortical effects aligns with contemporary hodotopic models emphasizing that speech deficits result from network disconnection rather than focal cortical damage. The external and internal capsules serve as critical hubs connecting frontal, temporal, and insular nodes, and their disruption affects multi-dimensional functional integration across the distributed language network. Although substantially larger effect sizes were observed in the LH, analyses including RH patients only demonstrated robust associations between damage to the external capsule, insula, and inferior fronto-occipital fasciculus and speech impairment severity. Moreover, patients with RH lesions tended to present with milder speech deficits despite larger lesion volumes, whereas patients with LH lesions were more likely to present with severe speech impairment even with smaller overall lesion volumes. This pattern is consistent with contemporary dual-stream models (left-dominant dorsal route; more bilateral ventral route) and prior acute lesion-symptom mapping studies examining pathway-specific aphasia phenotypes [6,7,8,12,60].

### 4.2. Primary Constraint: Clinical Neuroimaging and Aphasia Assessment

The interpretation and generalizability of our findings is inevitably constrained by the use of standard-of-care neuroimaging and a coarse measure of speech/language (NIHSS Best Language) obtained within 48 h of stroke onset. The Best Language subtest samples a narrow range of expressive/receptive functions (naming, reading, simple comprehension) and cannot disambiguate phonological, semantic, syntactic, prosodic, or pragmatic language deficits. Furthermore, it can be challenging to distinguish clinically meaningful speech deficits from non-linguistic factors that can halt speech production early after stroke. In acute settings, day-to-day fluctuations, edema, perfusion abnormalities, early diaschisis, medical complications, and fatigue can inflate or mask speech impairment and contaminate lesion-deficit inference [56,57,74,76]. In addition, interpreting acute lesion-symptom relationships can be challenging since acute lesion maps reflect not only structural injury but also transient network dysfunction, which limits strong claims about the neural architecture supporting language [57,76]. Consequently, some NIHSS-defined “mild speech deficit” cases may partly reflect transient speech disturbances rather than breakdowns in linguistic processes (cf. Ref. [74]).

Nevertheless, several important aspects of the study design combat these constraints. First, the NIHSS is psychometrically robust, widely implemented in clinical stroke care, and its prognostic value has been established in prior research [64,65,67,70,73]. Thus, LSM of the NIHSS captures the relationship between initial network vulnerability and acute speech deficits as assessed at the bedside [73,74,76]. The graded lesion profiles observed here (i.e., anatomical distinction for presence vs. severity of speech impairment in bilateral and left-dominant anatomy, respectively) align with the NIHSS’s empirically mapped deficit structure and predictive utility. Second, all analyses were adjusted for overall lesion volume, permutation-corrected to control for multiple comparisons, and our primary results were replicated across ROI-based and voxelwise approaches. The robust statistical approach minimizes the risk of spurious and artifactual effects. Finally, the lesion anatomy associated with presence and severity of speech impairment in LH patients is highly consistent with an extensive prior literature [9,10,12,57], suggesting that the equally consistent effects observed in RH patients accurately reflect acute lesion-symptom associations.

Therefore, an overview of our findings indicates that the right hemisphere contributes measurably to speech disruption when critical homolog nodes are damaged, even though the milder RH deficits may be transient in nature and do not necessarily herald chronic aphasia [1,58,61]. Although prior research has primarily focused on cortical areas of the RH “weak shadow” language network [28,92], our results suggest critical involvement of deep white matter structures and association pathways connecting temporoparietal and frontal areas. The fact that damage to this mirror network leads to acute speech deficits in a larger number of patients than prior estimates would suggest is consistent with the notion that the RH plays an important role in normal speech/language processing. In the same vein, our results challenge the historically left-centric emphasis in aphasia research and, instead, stipulate that future research should focus on elucidating what the right hemisphere can—as opposed to cannot—contribute to language recovery in aphasia.

These findings have several important clinical implications. First, clinicians should recognize that acute speech deficits following RH stroke are more common than traditionally appreciated. While these deficits tend to be milder and more transient than LH deficits, they represent genuine language disruption warranting speech-language evaluation. Second, the NIHSS Best Language subscore effectively identifies RH patients at risk for communication difficulties during the acute phase. Third, understanding that RH perisylvian and subcortical networks contribute to immediate speech function can inform prognostication and rehabilitation planning. Finally, given evidence that RH homolog regions participate in language processing, future research should investigate whether interventions targeting these regions (e.g., non-invasive brain stimulation) can enhance recovery outcomes in patients with LH damage.

### 4.3. Limitations

Several limitations may restrict the generalizability of these findings to other populations. As outlined above, the use of a coarse unidimensional measure of speech production and clinical neuroimaging constrains the extent to which our results inform concurrent neurobiological models of speech/language processing. Even so, the rigorous statistical approach lends confidence to the analyses as an accurate reflection of clinical reality. In addition, since handedness data was not available for the patients in this sample, we were unable to investigate the relationship between potential indices of pre-stroke language laterality and presence and severity of speech deficits after left vs. right hemisphere stroke. Left-handed patients are more likely to experience speech impairments after RH strokes [2], although the prevalence of crossed aphasia is comparatively low [93,94]. While it is reasonable to assume some degree of handedness confounding here, this effect is unlikely to account for the large number of patients presenting with speech deficits after damage to the RH. Finally, the cross-sectional study design limits our ability to delineate differences in recovery trajectories following left vs. right hemisphere lesions. Prior literature clearly demonstrates that chronic aphasia primarily affects LH stroke survivors, suggesting that most of the RH patients in the current study are likely to experience better recovery. Longitudinal studies combining comprehensive assessment batteries with structural and functional neuroimaging are needed to clarify the neuroplastic mechanisms underpinning language recovery in both patient groups.

## 5. Conclusions

Relying on a large sample of ischemic stroke patients, we found that lesion to subcortical white matter hubs and pathways in the right hemisphere was consistently associated with the presence and severity of acute speech deficits. The observed lesion-symptom associations were homologous to those observed in left hemisphere stroke, suggesting that damage to a mirror network in the right hemisphere leads to acute disruption of language function, albeit less severe than acute left hemisphere speech deficits. By defining acute lesion correlates in a large bilateral cohort, these findings advance our understanding of the neural architecture underlying language recovery and challenge unilateral models of post-stroke speech impairment.

## Figures and Tables

**Figure 1 brainsci-15-01329-f001:**
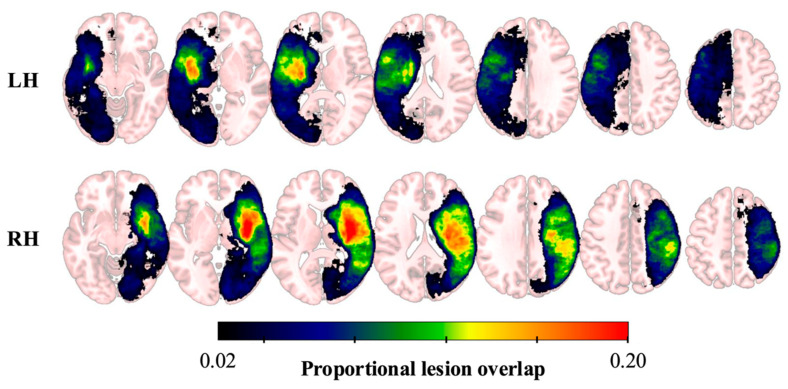
Lesion Overlap Maps for Participants with LH and RH Lesions. The colorbar indicates the proportion of overlapping lesions at each voxel, with warmer colors representing greater overlap.

**Figure 2 brainsci-15-01329-f002:**
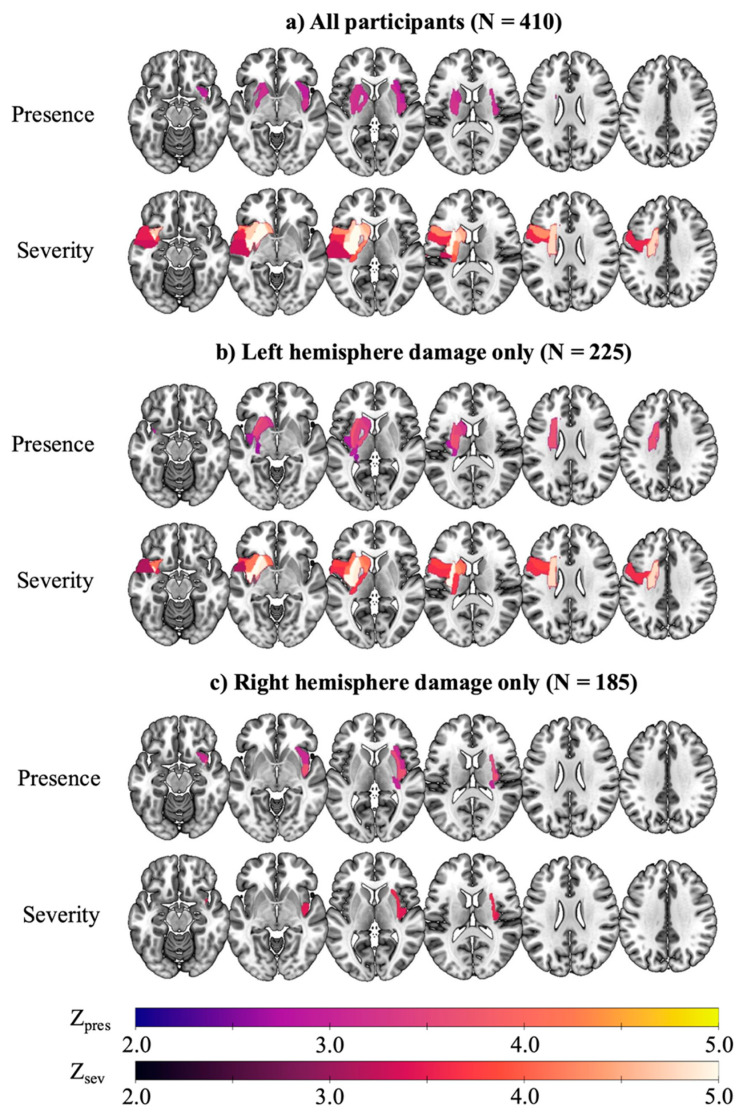
Region-of-Interest Lesion-Symptom Mapping of NIHSS Best Language Scores. Separate analyses were run for (**a**) the full sample (N = 410); (**b**) LH lesions only (N = 225); and, (**c**) RH lesions only (N = 185). Each analysis was run for (i) presence of speech deficits (NIHSS Best Language, 0 vs. ≥1) and (ii) severity of speech deficits (NIHSS Best Language score). Significant regions include ROIs lesioned in at least 10 participants that survived permutation thresholding (N = 5000) after adjusting for lesion volume. Z-scores are mapped from 2 to 5, with warmer colors indicating stronger associations between lesion presence and speech deficits.

**Figure 3 brainsci-15-01329-f003:**
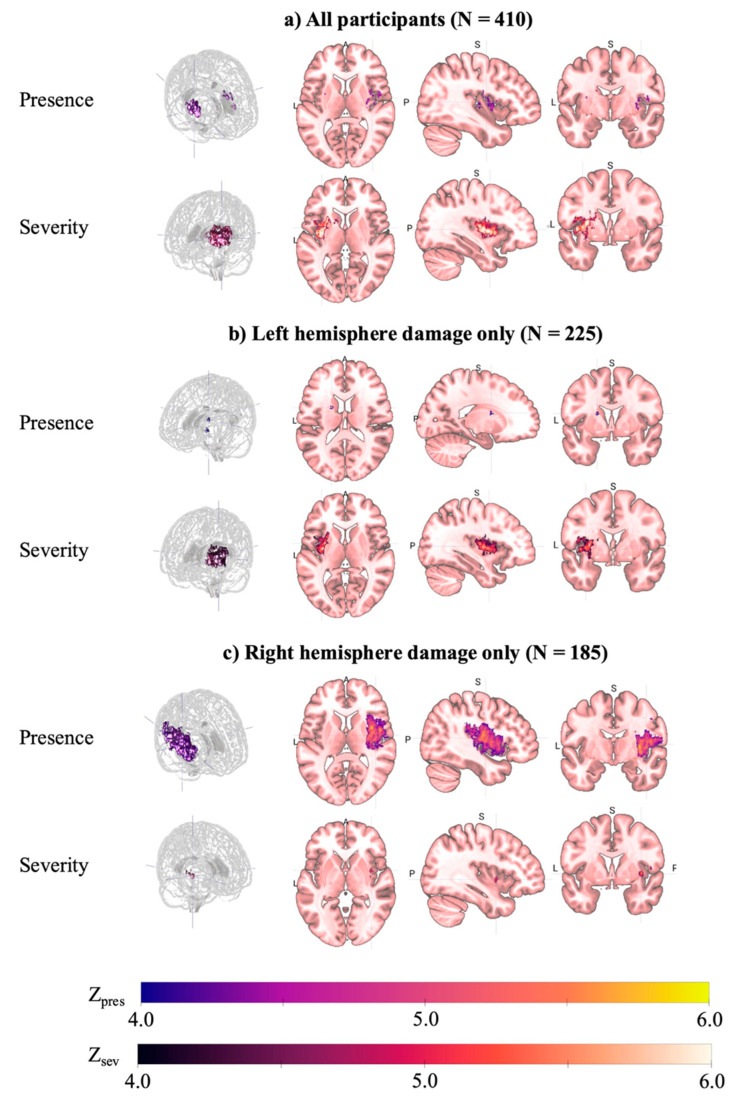
Voxelwise Lesion-Symptom Mapping of NIHSS Best Language Scores. Separate analyses were run for (**a**) the full sample (N = 410); (**b**) LH lesions only (N = 225); and, (**c**) RH lesions only (N = 185). Each analysis was run for (i) presence of speech deficits (NIHSS Best Language, 0 vs. ≥1) and (ii) severity of speech deficits (NIHSS Best Language score). Clusters represent significant lesion-deficit associations following permutation-based correction (*p* < 0.005; 5000 permutations), with lesion volume included as a covariate. Only voxels lesioned in ≥10 participants were included in analyses. Z-scores are mapped from 4 to 6, with warmer colors indicating stronger associations between lesion presence and speech deficits.

**Figure 4 brainsci-15-01329-f004:**
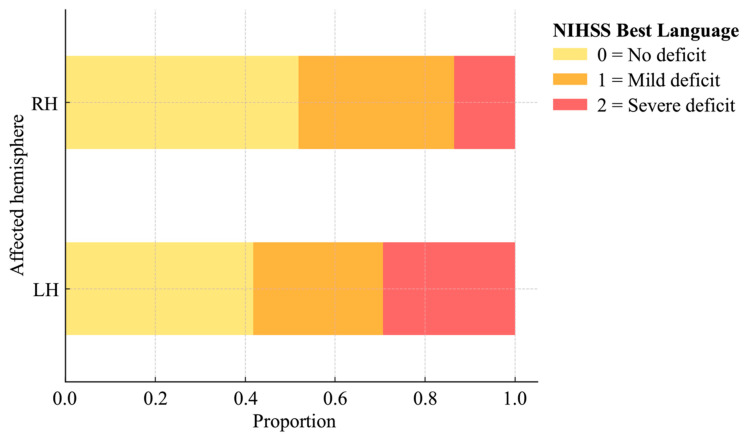
Distribution of NIHSS Best Language Scores by Hemisphere of Lesion Involvement. Speech deficits were observed in 58.2% of LH patients and 48.1% of RH patients. A significantly greater incidence of severe speech impairment (NIHSS Best Language score = 2) was observed in LH patients relative to RH patients (*p* < 0.001).

**Table 1 brainsci-15-01329-t001:** Study sample characteristics. Summary statistics for participants with vs. without speech deficits (NIHSS Best Language ≥ 1 vs. 0) following left or right hemisphere stroke. Continuous variables are presented as the mean and standard deviation, and discrete variables as counts. Group comparisons conducted using independent t-tests for continuous variables and chi-square tests for categorical variables. Abbreviations: AA = African American, cc = cubic centimeters, F = female, LL = lesion load, M = male, NIHSS = National Institute of Health Stroke Scale, W = White, y = years. * *p* < 0.05, ** *p* < 0.01.

Variable	No Deficit	Speech Deficit	*p*
	**Left Hemisphere Damage (N = 225)**		
N	94	131	
LesVol, cc	29.9 ± 85.0	39.8 ± 74.5	0.364
LH_LL, %	0.02 ± 0.05	0.05 ± 0.08	0.004 **
RH_LL, %	<0.01 ± 0.03	<0.01 ± 0.01	0.573
Age, y	65.0 ± 68.0	68.0 ± 14.7	0.123
NIHSS	4.5 ± 5.5	12.2 ± 9.1	<0.001 **
Sex, F/M	51/43	67/62	0.836
Race, AA/W	24/65	31/94	0.842
	**Right hemisphere damage (N = 185)**		
N	96	89	
LesVol, cc	24.7 ± 48.8	61.6 ± 74.1	<0.001 **
LH_LL, %	<0.01 ± 0.01	<0.01 ± 0.01	0.134
RH_LL, %	0.02 ± 0.05	0.06 ± 0.07	<0.001 **
Age, y	63.3 ± 15.1	65.9 ± 14.7	0.229
NIHSS	3.6 ± 5.1	11.3 ± 7.6	<0.001 **
Sex, F/M	48/48	46/43	0.935
Race, AA/W	15/76	27/59	0.020 *

## Data Availability

The anonymized images for the Stroke Outcome Optimization Project are available from OpenNeuro (https://openneuro.org/datasets/ds004889, accessed on 6 January 2025). Behavioral data are available from the corresponding author upon reasonable request.

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
