# Peer review of "Brain Sci.2025, 15(12), 1329;https://doi.org/10.3390/brainsci15121329"

_brainsci, 2025, doi:10.3390/brainsci15121329_

Round 1

Reviewer 1 Report

Comments and Suggestions for Authors

GENERAL COMMENTS

This manuscript reports a post-hoc evaluation of right-hemisphere lesions and their importance for speech deficits. The manuscript re-analyzes imaging data from a large sample of stroke patients and concludes that right-hemisphere damage can and does result in speech deficits, challenging a left-centric view of language in the brain.

I believe the topic and data presented in the manuscript to be worthwhile for publication, however I have some concerns about emphasizing the novelty of the paper and some methodological clarifications I would like the authors to include prior to publication. I would like to preface that I am not an expert in imaging and language localization, but I do work in aphasia with various complex methodologies. I therefore do not focus on the localization results in my review.

SPECIFIC COMMENTS:

  • My main issue with the paper as it stands is that it is not obvious what the paper adds to the existing body of literature. The authors rightly cite past research including both healthy and clinical groups that suggests the right hemisphere is actively involved in language processing (e.g. in lines 72-90). It would therefore follow logically that RH damage could lead to language deficits. Overall I think the authors should greatly emphasise why this study is needed, what novelty it brings, and what knowledge will remain unknown without this paper. I would recommend not just working this into lines 72-90 but throughout the introduction and discussion.
  • Table 1: This table is fairly confusing and could do with reorganization. My main issue here though is that race is binarized to W/AA: were participants of no other races included in the study? Or were those of other races lumped in with W or AA? Either way I suspect there would be a potentially significant issue there.
  • It is also unclear to me how a distinction was made between participants with and without a speech deficit, and why this is a binary distinction. Was the NIHSS used to distinguish participants with and without deficits? That seems obvious to me but it might not be the: if, as the authors say in lines 173-4, the NIHSS had a continuous outcome scale of 0-42, where was the cut-off placed? Also, in lines 313-314 they state that NIHSS scores were separate from the deficit/no deficit distinction. This requires explaining.
  • The authors should include some details of the scoring of the Best Language subtest in lines 188-197. If this is data they obtained post-hoc, which seems to be the case, they should include as much detail as possible on the original scoring.
  • Related to this, if the Best Language subtest ranges from 0 to 3, would this not drastically limit the possible ranges of discovered deficits? If only integer responses of 0, 1, 2, and 3 are in the resulting data this presents a serious statistical problem. Please explain further.
  • This also relates to the authors' mention of their collating scores of 2 and 3 (246-7). This may suggest Best Language was not used as a continuous but rather a categorical predictor?
  • Lines 271-280: I would encourage the authors to reframe this paragraph around lateralization as well. As it stands I do not think it adds anything new above what the existing literature on lesion localization has found.
  • In line 289, please re-state that severity is represented by Best Language score.
  • Line 313-4 and 339-341: please see my earlier comment on how the presence of speech deficits was determined. This is a crucial issue for this manuscript.
  • The first paragraph of the discussion should again include a mention of why this paper is necessary and novel. As it stands I am not convinced this manuscript addresses an actual research gap so the authors should state this far more explicitly.
  • I do think the discussion is fairly strong as it stands as long as the study's novelty is properly addressed. However the authors might want to include a paragraph or even just a few sentences on what these findings mean clinically: what can clinical scientists or clinicians glean from these findings? What changes for aphasia assessment and treatment now that we know the RH is involved in this way?

Author Response

Reviewer 1

Comments and Suggestions for Authors
This manuscript reports a post-hoc evaluation of right-hemisphere lesions and their importance for speech deficits. The manuscript re-analyzes imaging data from a large sample of stroke patients and concludes that right-hemisphere damage can and does result in speech deficits, challenging a left-centric view of language in the brain.

I believe the topic and data presented in the manuscript to be worthwhile for publication, however I have some concerns about emphasizing the novelty of the paper and some methodological clarifications I would like the authors to include prior to publication. I would like to preface that I am not an expert in imaging and language localization, but I do work in aphasia with various complex methodologies. I therefore do not focus on the localization results in my review.

Authors’ response: We thank the reviewer for their enthusiasm for our work and the constructive feedback provided. We have responded to their feedback, and we believe these edits have substantially strengthened our manuscript.  

Specific Comments

Comment 1: My main issue with the paper as it stands is that it is not obvious what the paper adds to the existing body of literature. The authors rightly cite past research including both healthy and clinical groups that suggests the right hemisphere is actively involved in language processing (e.g. in lines 72-90). It would therefore follow logically that RH damage could lead to language deficits. Overall I think the authors should greatly emphasise why this study is needed, what novelty it brings, and what knowledge will remain unknown without this paper. I would recommend not just working this into lines 72-90 but throughout the introduction and discussion.

Authors’ response: We thank the reviewer for this important comment. We have revised the manuscript to more clearly articulate the novelty and significance of our contribution. While prior research has indeed demonstrated that the right hemisphere is involved in language processing in healthy individuals, few studies have systematically investigated the acute effects of lesions on speech deficits using large-scale lesion-symptom mapping. This is perhaps primarily a reflection of the clinical reality of acute stroke management, where the emphasis is placed on lifesaving procedures and stabilizing patients’ medical conditions. Thus, the key gaps our study addresses are: (1) most prior RH studies recruited small samples or excluded RH patients entirely, limiting generalizability; (2) existing work has primarily focused on chronic stroke survivors, where time-dependent reorganization within the intact LH network may mask more subtle lesion-symptom associations in the RH; (3) the regional distribution of lesion anatomy specifically associated with acute speech deficits after RH stroke remains incompletely characterized due to these limitations.

The novelty of our study lies in examining a large, unselected cohort (N=410; 185 RH patients) assessed within 48 hours of stroke onset. This enables us to characterize the immediate contribution of RH regions to speech function before compensatory reorganization occurs. Our study represents one of the largest acute lesion-symptom mapping studies of RH speech deficits to date, which provides adequate statistical power to identify bilateral lesion-symptom associations. In the revised manuscript, we have strengthened the rationale throughout the Introduction (pp. 3-5) and Discussion (pp. 15-18) to emphasize these unique contributions and their implications for understanding bilateral language representation and neuroplastic recovery mechanisms.

Comment 2: Table 1: This table is fairly confusing and could do with reorganization. My main issue here though is that race is binarized to W/AA: were participants of no other races included in the study? Or were those of other races lumped in with W or AA? Either way I suspect there would be a potentially significant issue there.

Authors’ response: We have revised Table 1 for improved clarity by re-organizing LH vs. RH damage groups vertically as opposed to horizontally, and by including more detailed caption explaining the statistical comparisons. Regarding race, the SOOP dataset includes only African-American (AA) and White (W) participants; no participants of other racial backgrounds were enrolled during the study period (January 2019-December 2020) or their race was not reported in the dataset. This demographic distribution broadly reflects the patient population served by the Prisma Health-Upstate hospital in Greenville, South Carolina. While we acknowledge this limitation, we wish to emphasize that investigating racial differences in bilateral lesion distribution is beyond the scope of this work.

Comment 3: It is also unclear to me how a distinction was made between participants with and without a speech deficit, and why this is a binary distinction. Was the NIHSS used to distinguish participants with and without deficits? That seems obvious to me but it might not be the: if, as the authors say in lines 173-4, the NIHSS had a continuous outcome scale of 0-42, where was the cut-off placed? Also, in lines 313-314 they state that NIHSS scores were separate from the deficit/no deficit distinction. This requires explaining.

Authors’ response: We appreciate the reviewer’s request for clarification. The distinction between participants with and without speech deficits was based solely on the NIHSS Best Language subscore, not the total NIHSS score (range, 0-42 points). The Best Language subtest has its own 0-3 scoring scale, where: 0 = no aphasia, 1 = mild-moderate aphasia, 2 = severe aphasia, and 3 = mute/global aphasia. We operationalized “presence of speech deficits” as any score ≥1 (indicating some degree of language impairment) versus a score of 0 (indicating no detectable language impairment), and “severity of speech deficits” was defined based on integer scores of 0, 1, or 2/3. The primary data analysis relating lesion damage to these behavioral domains was run separately for presence and severity of speech deficits. Since this was a retrospective study based on clinical data, including assessment of stroke severity (NIHSS) and clinical intake neuroimaging, these data reflect the clinical reality of acute stroke management. Thus, although the NIHSS Best Language is a coarse assessment of speech deficits, our data analysis plan was designed to maximize its utility for lesion-symptom mapping (i.e., running the analyses on binarized and hierarchical behavioral measures), and we have made a point of appropriately acknowledging the limited generalizability of our findings in the context of more detailed language assessments. In addition to the Best Language subscore, which served as the dependent variable in all primary analyses, the total NIHSS score is reported separately as a measure of overall stroke severity for descriptive purposes. We have clarified this throughout the manuscript (e.g., first sentence of Data Analysis, p. 8) to eliminate confusion between the Best Language subscore and the total NIHSS score.

Comment 4: The authors should include some details of the scoring of the Best Language subtest in lines 188-197. If this is data they obtained post-hoc, which seems to be the case, they should include as much detail as possible on the original scoring.

Authors’ response: We have expanded the description of the Best Language scoring procedures to provide greater methodological transparency (pp. 6-7). The NIHSS Best Language was administered by a trained stroke neurologist within 48 hours of hospital admission, following standardized NIHSS administration protocols. Our study analyzed these scores retrospectively from the SOOP dataset. The scoring involves structured assessment of naming (6 common objects), reading (simple sentence), and comprehension (following verbal commands), with clinicians rating overall language function on a 0-3 scale based on predetermined criteria. The “Assessment of Speech Deficits” section of the revised manuscript provides a detailed description of: (1) the specific tasks administered, (2) the scoring criteria for each level (0-3), and (3) the scoring procedures. We hope these details clarify that while our analysis is retrospective, the original NIHSS data were collected prospectively using rigorous, standardized clinical protocols.

Comment 5: Related to this, if the Best Language subtest ranges from 0 to 3, would this not drastically limit the possible ranges of discovered deficits? If only integer responses of 0, 1, 2, and 3 are in the resulting data this presents a serious statistical problem. Please explain further.

Authors’ response: The reviewer raises an important methodological point. The NIHSS Best Language subscore is an ordinal scale with only four possible values (0, 1, 2, 3), which limits granularity compared to comprehensive aphasia batteries. We acknowledge this as a significant constraint in the revised manuscript (p. 7; p. 17). However, several factors mitigate this limitation for our specific research question: (1) The large sample size (N=410) provides sufficient statistical power to detect associations even with coarse measurement; (2) We employed permutation-based statistical thresholding (5,000 iterations) specifically designed to handle non-parametric, ordinal data; (3) We analyzed both binary presence/absence (0 vs. ≥1) and ordinal severity (0, 1, 2-3), capturing both categorical and graded effects; (4) The NIHSS Best Language subtest has demonstrated psychometric validity and predictive utility in prior research (references cited on p. 7); (5) Our focus on acute lesion-symptom associations (identifying which brain regions contribute to immediate speech disruption) is appropriately matched to this level of measurement, whereas more detailed linguistic characterization might be confounded by acute medical factors. We have outlined these issues in the Limitations section to address the ordinal nature of the outcome measure and its implications for interpreting our findings.

Comment 6: This also relates to the authors' mention of their collating scores of 2 and 3 (246-7). This may suggest Best Language was not used as a continuous but rather a categorical predictor?

Authors’ response: The reviewer is correct. The NIHSS Best Language subscore was treated as an ordinal (not continuous) variable in our statistical analyses, and all data analyses were designed accordingly. Given the discrete nature of the 0-3 scale and the small number of participants with scores of 3 (N=6), we collapsed scores 2 and 3 into a single “severe” category, creating a three-level ordinal outcome: 0 (no deficit), 1 (mild), and 2-3 (severe). This approach is statistically appropriate for ordinal data and has been used in prior NIHSS-based lesion-symptom mapping studies. Our permutation-based statistical methods (NiiStat software) are specifically designed to handle ordinal outcomes and do not assume continuous, normally distributed data. We have clarified this throughout the Methods and Results sections to explicitly state that severity was analyzed as an ordinal variable, and we provide justification for collapsing the two highest scores.

Comment 7: Lines 271-280: I would encourage the authors to reframe this paragraph around lateralization as well. As it stands I do not think it adds anything new above what the existing literature on lesion localization has found.

Authors’ response: We thank the reviewer for asking for clarification about how these results inform the language lateralization literature. To recap briefly, we conducted whole-brain and hemisphere-specific analyses separately to probe lesion-symptom associations in the full unselective sample of acute ischemic stroke survivors (whole-brain) and in selected subsamples of participants with damage affecting only one hemisphere and not the other (hemisphere-specific). The purpose of the first set of analyses was to identify lesion anatomy associated with speech deficits irrespective of the affected hemisphere. Although we would expect to observe left-dominant effects given the critical role of the left hemisphere in language processing, an effect in the right hemisphere would need to be robust enough to survive permutation-based thresholding. Therefore, it serves as a stringent test of the relationship between right hemisphere lesion and acute speech deficits. The second set of analyses was designed to examine the spatial distribution of lesion anatomy associated with speech deficits specifically in individuals with left vs. right hemisphere damage. We have focused on describing these analyses in detail in the revised manuscript.

Lines 271-280 describe the voxelwise and region-of-interest whole-brain lesion-symptom mapping results. We found that the presence of speech deficits (NIHSS Best Language ≥1) was associated with lesion affecting a bilateral network of subcortical, white matter, and insular structures. These results are described in the Results, Figure 2, and Supplementary Table 1. The implications of the bilateral effects in the context of these analyses are discussed in the Discussion section (e.g., “Presence vs. Severity of Speech Deficits”). 

Comment 8: In line 289, please re-state that severity is represented by Best Language score.

Authors’ response: We have added this clarification at line 289 and clarified conceptual ambiguity throughout the Results and Discussion sections to ensure readers consistently understand that severity refers specifically to the ordinal NIHSS Best Language score. 

Comment 9: Line 313-4 and 339-341: please see my earlier comment on how the presence of speech deficits was determined. This is a crucial issue for this manuscript.

Authors’ response: As clarified in our response to Comment 3, presence of speech deficits was determined solely by NIHSS Best Language subscore ≥1 (vs. 0). We have now consistently used this terminology throughout the Results section and added clarifying language wherever the distinction between presence/absence might be ambiguous.

Comment 10: The first paragraph of the discussion should again include a mention of why this paper is necessary and novel. As it stands I am not convinced this manuscript addresses an actual research gap so the authors should state this far more explicitly.

Authors’ response: We have revised the opening paragraph of the Discussion to foreground the novelty and necessity of this work. The revised paragraph explicitly states: (1) This is one of the largest acute LSM studies of RH speech deficits (N=185 RH patients) to date; (2) Assessment within 48 hours captures immediate contributions before compensatory reorganization; (3) Prior work has suffered from small samples, exclusion of RH patients, or focus on chronic rather than acute phases; (4) Our findings demonstrate that RH damage consistently produces acute speech deficits to a greater extent than previously thought; and, (5) The regional distribution of acute RH lesion-symptom associations provides novel insights into bilateral language architecture. This reframing makes clear that our study fills a critical empirical gap and challenges prevailing assumptions about hemispheric contributions to language.

Comment 11: I do think the discussion is fairly strong as it stands as long as the study's novelty is properly addressed. However the authors might want to include a paragraph or even just a few sentences on what these findings mean clinically: what can clinical scientists or clinicians glean from these findings? What changes for aphasia assessment and treatment now that we know the RH is involved in this way?

Authors’ response: We thank the reviewer for this important suggestion. We have added a new paragraph right before the “Limitations” section explicitly discussing clinical implications. Key clinical takeaways include: (1) Clinicians should recognize that acute speech deficits after RH stroke are common and represent genuine language disruption, not solely visuospatial or attention deficits; (2) The NIHSS Best Language subscore effectively identifies RH patients at risk for communication difficulties; (3) Even mild acute RH deficits warrant speech-language evaluation and monitoring, as they may impact communication effectiveness despite not meeting criteria for classic aphasia; (4) Understanding bilateral language architecture can inform prognostication as RH patients typically show better recovery but may benefit from targeted intervention during acute phase; (5) Future rehabilitation research should investigate whether interventions targeting RH homolog regions can enhance recovery in LH stroke patients, given evidence that these regions contribute to language function. These clinical implications bridge our basic science findings to bedside practice and rehabilitation research.

Reviewer 2 Report

Comments and Suggestions for Authors

The manuscript presents interesting findings and is generally well organized. However, several points need to be addressed during manuscript revision.  

  1. The introduction provides a comprehensive and well-referenced background on hemispheric contributions to post-stroke speech deficits. However, it is overly long and can be improved by reducing redundancy, condensing historical context, and emphasizing the main research gap earlier to enhance clarity and focus.
  2. The authors report that larger lesion volumes are associated with greater speech deficits. Consider including a correlation graph to illustrate the relationships between lesion volume, overall NIHSS score, and speech deficit severity.
  3. The authors report higher speech impairment among African-American patients with RH lesions. Could this difference reflect confounding factors such as lesion volume or stroke subtype? Clarification or adjusted analysis would improve interpretation.
  4. The authors identify strong associations between speech deficits and deep subcortical white matter lesions. Could they clarify whether cortical regions (e.g., inferior frontal gyrus or superior temporal gyrus) also contributed independently after controlling for lesion volume?
  5. Please label the figure panels as (a), (b), (c), etc., and include corresponding descriptions in the figure legends for clarity.
  6. The discussion highlights structural correlates but provides limited insight into clinical relevance. How might these findings inform early clinical assessment or rehabilitation strategies for patients with right hemisphere stroke presenting with speech deficits?
  7. Given its coarse sensitivity, do the authors anticipate that using more detailed language batteries (e.g., WAB, BDAE) in future studies might reveal subtler or domain-specific effects of right hemisphere lesions?
  8. The findings highlight involvement of deep white matter and association pathways, but the functional implications could be more deeply explored. Do the authors propose that disruption of interhemispheric or fronto-temporal connectivity contributes to acute deficits beyond focal cortical injury?
  9. Clarify whether lesion–symptom mapping results were adjusted for potential confounds such as age or stroke type.

Author Response

Reviewer 2

Comments and Suggestions for Authors
The manuscript presents interesting findings and is generally well organized. However, several points need to be addressed during manuscript revision.  

Authors’ response: We thank the reviewer for their enthusiasm for our work and the constructive feedback provided. We have responded to their feedback, and we believe these edits have substantially strengthened our manuscript.  

Specific Comments

Comment 1: The introduction provides a comprehensive and well-referenced background on hemispheric contributions to post-stroke speech deficits. However, it is overly long and can be improved by reducing redundancy, condensing historical context, and emphasizing the main research gap earlier to enhance clarity and focus.

Authors’ response: We appreciate this constructive feedback and have condensed the Introduction while maintaining essential background information. Specific revisions include: (1) Streamlined the historical overview by removing redundant examples and focusing on the most influential cases; (2) Consolidated discussion of dual-stream models to eliminate repetition; (3) Emphasized the primary research gap and study rationale early on so readers immediately gauge the study’s purpose; (4) Reduced the paragraph on compensatory RH involvement by condensing contradictory findings into a more focused synthesis. The revised Introduction maintains comprehensive coverage while improving flow and emphasizing our novel contributions more prominently.

Comment 2: The authors report that larger lesion volumes are associated with greater speech deficits. Consider including a correlation graph to illustrate the relationships between lesion volume, overall NIHSS score, and speech deficit severity.

Authors’ response: We thank the reviewer for this suggestion. We have added a supplementary figure (Supplementary Figure 1) displaying a Spearman’s correlation heatmap between lesion volume, NIHSS Best Language scores, age, and NIHSS scores. The correlation results show a significant correlation between lesion volume and NIHSS Best Language scores (rho=0.30, p<0.01) and NIHSS scores (rho=0.48, p<0.01), as well as between age and NIHSS Best Language scores (rho=0.11, p<0.05). These results emphasize the importance of adjusting all primary analyses for the effect of overall lesion volume. We reference this supplementary figure in the Results section and discuss its implications.

Comment 3: The authors report higher speech impairment among African-American patients with RH lesions. Could this difference reflect confounding factors such as lesion volume or stroke subtype? Clarification or adjusted analysis would improve interpretation.

Authors’ response: The reviewer is correct: The incidence of speech deficits was greater among African-American than White individuals after RH stroke. Post hoc analyses revealed no significant difference in lesion volume (AA/W: 37.4 vs. 36.5 cc, t=0.09, p=0.931) or NIHSS scores (AA/W: 8.10 vs. 6.19, t=1.40, p=0.168); however, we did observe a statistically significant difference in age (AA/W: 59.0 vs. 68.5 years, t=3.53, p<0.01). This is not an unusual pattern of findings, and likely reflects compounding effects of racial health disparities, including poorer access to healthcare and, subsequently, worse health outcomes. However, since the effects of race on the incidence of speech deficits in left vs. right hemisphere stroke survivors was beyond the scope of this work, these results are not emphasized in the revised manuscript.

Comment 4: The authors identify strong associations between speech deficits and deep subcortical white matter lesions. Could they clarify whether cortical regions (e.g., inferior frontal gyrus or superior temporal gyrus) also contributed independently after controlling for lesion volume?

Authors’ response: This is an important question regarding the independent contributions of cortical vs. subcortical structures. Our ROI-based analyses, which were adjusted for overall lesion volume, did identify significant independent effects in cortical regions, including the inferior frontal gyrus pars opercularis (Z=4.30, p<0.05) and superior temporal gyrus (Z=3.31, p<0.05) for LH lesions. However, the strongest effects were indeed in subcortical structures (external capsule Z=6.39; posterior insula Z=5.64), suggesting that white matter disconnection and subcortical damage contribute more substantially to acute speech deficits than isolated cortical injury. This pattern aligns with the “hodotopic” view that network disconnection, rather than focal cortical damage alone, drives acute speech deficits. We have clarified this in the Discussion, explicitly stating that while cortical regions contribute independently, subcortical pathways show stronger associations, likely reflecting their role as critical hubs connecting distributed language networks.

Comment 5: Please label the figure panels as (a), (b), (c), etc., and include corresponding descriptions in the figure legends for clarity.

Authors’ response: We have revised all figures including panels with relevant labels ((a), (b), (c)), and expanded the figure legends to explicitly describe each panel. Figures 2 and 3 now clearly indicate panels for (a) full sample, (b) left hemisphere damage, and (c) right hemisphere damage. This organization substantially improves figure readability and helps readers navigate the multiple analyses presented.

Comment 6: The discussion highlights structural correlates but provides limited insight into clinical relevance. How might these findings inform early clinical assessment or rehabilitation strategies for patients with right hemisphere stroke presenting with speech deficits?

Authors’ response: We have added a dedicated paragraph on clinical implications before the “Limitations” section of the Discussion, also addressing Reviewer 1’s comment. Key clinical takeaways include: (1) Clinicians should recognize that acute speech deficits after RH stroke are common and represent genuine language disruption, not solely visuospatial or attention deficits; (2) The NIHSS Best Language subscore effectively identifies RH patients at risk for communication difficulties; (3) Even mild acute RH deficits warrant speech-language evaluation and monitoring, as they may impact communication effectiveness despite not meeting criteria for classic aphasia; (4) Understanding bilateral language architecture can inform prognostication as RH patients typically show better recovery but may benefit from targeted intervention during acute phase; (5) Future rehabilitation research should investigate whether interventions targeting RH homolog regions can enhance recovery in LH stroke patients, given evidence that these regions contribute to language function. These clinical implications bridge our basic science findings to bedside practice and rehabilitation research.

Comment 7: Given its coarse sensitivity, do the authors anticipate that using more detailed language batteries (e.g., WAB, BDAE) in future studies might reveal subtler or domain-specific effects of right hemisphere lesions?

Authors’ response: This is an excellent point for future research directions. We fully anticipate that comprehensive aphasia batteries like the Western Aphasia Battery (WAB) or Boston Diagnostic Aphasia Examination (BDAE) would reveal more nuanced, domain-specific effects of RH lesions that our coarse measure could not detect. For example, detailed assessment might identify selective impairments in prosodic processing, discourse coherence, figurative language comprehension, or pragmatic communication, all of which are domains where RH contributions may be particularly important. However, such detailed assessment in the acute phase (within 48 hours) faces practical challenges: patients may be medically unstable, fatigued, or unable to tolerate lengthy testing. The NIHSS Best Language subscore, while limited, offers the advantage of feasibility and standardization in acute settings. We have expanded the Discussion and Limitation sections to recommend that future studies conduct comprehensive evaluation at subacute timepoints when patients are more stable, enabling fine-grained characterization of RH language deficits across the recovery trajectory.

Comment 8: The findings highlight involvement of deep white matter and association pathways, but the functional implications could be more deeply explored. Do the authors propose that disruption of interhemispheric or fronto-temporal connectivity contributes to acute deficits beyond focal cortical injury?

Authors’ response: This is a highly insightful question that touches on fundamental debates in cognitive neuroscience regarding network vs. localizationist views of language. We have expanded the Discussion to more thoroughly explore the functional significance of white matter disruption. Briefly, we propose that acute speech deficits result primarily from network disconnection rather than focal cortical damage; a “hodotopic” rather than purely “localizationist” framework. Specifically: (1) The external capsule contains extreme capsule fibers forming part of the ventral language pathway, connecting temporal semantic regions with frontal production areas; it’s disruption disconnects critical network nodes; (2) The internal capsule carries thalamocortical projections that modulate cortical activity and support speech motor control; (3) The fronto-occipital fasciculi enable long-range integration between frontal executive regions and posterior semantic/phonological areas. Our findings show that these pathways, bilaterally, are more strongly associated with speech deficits than isolated cortical damage, supporting a network disconnection model. Moreover, the bilateral involvement of homologous pathways suggests that speech function depends on coordinated interhemispheric communication, where disruption of either hemisphere’s contribution compromises the integrated network.

Comment 9: Clarify whether lesion–symptom mapping results were adjusted for potential confounds such as age or stroke type.

Authors’ response: We thank the reviewer for requiring clarification on the LSM approach. As described in the Data Analysis section, all LSM analyses controlled for total lesion volume as a covariate in order to adjust for the expected relationship between lesion size and speech deficits. Moreover, stroke type was inherently controlled as all participants in the SOOP dataset presented with acute ischemic stroke. However, we did not adjust the analyses for participants’ age as investigating age-related variance in lesion anatomy was beyond the scope of this work. We have aimed to clarify all relevant aspects of the analytical approach in the revised manuscript.  

Round 2

Reviewer 2 Report

Comments and Suggestions for Authors

The author successfully addressed all the queries and updated the manuscript accordingly. I recommend the manuscript for publication.